# Revealing the spatial nature of sublattice symmetry

**Rong Xiao[1] & Y. X. Zhao [2] ✉**

The sublattice symmetry on a bipartite lattice is commonly regarded as the chiral symmetry in the AIII class of the tenfold Altland–Zirnbauer classification. Here, we reveal the spatial nature of sublattice symmetry and show that this assertion holds only if the periodicity of primitive unit cells agrees with that of the sublattice labeling. In cases where the periodicity does not agree, sublattice symmetry is represented as a glide reflection in energy–momentum space, which inverts energy and simultaneously translates some $k$ by $\pi$, leading to substantially different physics. Particularly, it introduces novel constraints on zero modes in semimetals and completely alters the classification table of topological insulators compared to class AIII. Notably, the dimensions corresponding to trivial and nontrivial classifications are switched, and the nontrivial classification becomes $\mathbb{Z}_2$ instead of $\mathbb{Z}$. We have applied these results to several models, including the Hofstadter model both with and without dimerization.

Symmetry-protected topological band theory has been one of the main focuses of condensed matter research for around two decades[1–3]. For the tenfold Altland–Zirnbauer symmetry classes involving time-reversal symmetry ($T$), particle-hole symmetry ($C$), and chiral symmetry ($\Pi$), various periodic topological classification tables have been produced[4–14], which played a seminal role in organizing and discovering novel topological phases.

The original paper by Altland and Zirnbauer concerned the Bogoliubov–de Gennes (BdG) Hamiltonians for superconductors[4]. In this context, $\Pi$ was naturally introduced as a combination of $T$ and $C$, namely $\Pi = CT$ for the algebraic completeness. More strictly, $\Pi = i^s CT$, where $s = 0, 1$ and the front coefficient $i^s$ is assigned to ensure the convention $(\Pi)^2 = 1$. For the one-particle Hamiltonian $H$, the particle-hole symmetry $C$ anti-commutes with $H$, so the chiral symmetry $\Pi$ is a unitary operator that anti-commutes with the one-particle Hamiltonian $H$,

$$\{H, \Pi\} = 0. \tag{1}$$

Meanwhile, on a bipartite lattice consisting of two equal sublattices A and B, the sublattice symmetry assigns $\pm 1$ for $A$-sites and $B$-sites, respectively, and therefore can be represented as $S = \mathrm{diag}(1_A, -1_B)$ (see

Fig. 1). If hopping occurs only from one sublattice to the other, the sublattice symmetry $S$ anti-commutes with the one-particle Hamiltonian $H$ as well

$$\{H, S\} = 0. \tag{2}$$

Hence, the sublattice symmetry $S$ and chiral symmetry $\Pi$ usually are not distinguished in the literature[3–5,7–14], and the two terms, "sublattice" and "chiral", have been used interchangeably. Furthermore, the topological classification for sublattice symmetry is commonly believed as a completely solved problem, since it is understood that the topological classification tables of chiral symmetry can be directly applied to crystals with sublattice symmetry.

In this work, we reveal an essential difference between chiral symmetry $\Pi$ and sublattice symmetry $S$. In the BdG Hamiltonian, $\Pi$ as the combination of $T$ and $C$ is an internal symmetry, since $T$ and $C$ are both internal symmetries. However, generically $A$ and $B$ sublattices are spatially separate on a bipartite lattice (see the two simple examples in Fig. 1), and therefore the sublattice symmetry $S$ has an inherent spatial nature.

Considering the spatial nature, we can classify sublattice symmetries into two categories. If the periodicity of the primitive unit cells

[1]National Laboratory of Solid State Microstructures and Department of Physics, Nanjing University, Nanjing 210093, China. [2]Department of Physics and HK Institute of Quantum Science & Technology, The University of Hong Kong, Pokfulam Road, Hong Kong, China. ✉e-mail: yuxinphy@hku.hk

agrees with that of the $A$–$B$ labeling of lattice sites, the sublattice symmetry falls into class I, as demonstrated by the Su–Schrieffer–Heeger (SSH) model in Fig. 1a[15]. Otherwise, it falls into class II, where the 1D model with uniform nearest neighbor hoppings in Fig. 1b serves as a simple example.

Only class-I sublattice symmetry adheres to the theory of chiral symmetry, while class-II sublattice symmetry is represented as a glide reflection in energy–momentum space, leading to completely different topological physics.

There are two scenarios to consider for class-II sublattice symmetry. In the first scenario, each primitive unit cell comprises an odd number of states. The symmetry constraint results in $4n + 2$ zero modes with $n = 0, 1, 2, \cdots$, meaning that the minimum number of zero modes is two. Therefore, to achieve an insulator, we need to examine the second scenario where each primitive unit cell contains an even number of states. In this case, the number of zero modes is $4n$ with $n = 0, 1, 2, \cdots$.

Consequently, the topological classification table differs significantly from that of chiral symmetry. It becomes nontrivial in even dimensions and trivial in odd dimensions, whereas the table for chiral symmetry is nontrivial in odd dimensions and trivial in even dimensions. The nontrivial classification is now given as $\mathbb{Z}_2$ rather than $\mathbb{Z}$ for chiral symmetry.

Sublattice symmetry is a pervasive feature observed in various quantum materials[16–21], as well as in appropriately designed artificial crystals such as photonic and acoustic crystals, periodic mechanical systems, cold atoms in optical lattices, and periodic-electric arrays[22–35]. Thus, our work not only enhances our understanding of a fundamental aspect of sublattice symmetry-protected topology but also holds significant potential for wide applications in topological physics.

## RESULTS

### Two classes of sublattice symmetries

Let us start with enunciating the two classes of sublattice symmetries.

For a lattice model, once the primitive unit cell is chosen, the translation symmetry is described by unit translation operator $L_i$, which maps each unit cell to its neighbors. If the translation symmetry is also preserved by the sublattice bipartition, translation operators commute with the sublattice operator $[L_i, S] = 0$. Then, the sublattice symmetry is in class I with each unit cell having the same $A$-$B$ labeling of states (see Fig. 1a). Class-I $S$ can be effectively regarded as an internal symmetry in the $k$ space due to $[L_i, S] = 0$, and therefore adheres to all topological classifications for chiral symmetry $\Pi$.

Meanwhile, it is also ubiquitous that some translation operators exchange $A$ and $B$ sublattices. Then, a unit cell and its neighbor related by such a translation operator have opposite $A$–$B$ labeling (see Fig. 1b), and the sublattice symmetry is in class II. For class-II $S$, at least one translation operator, say $L_x$, anti-commutes with $S$,

$$\{L_x, S\} = 0. \tag{3}$$

This is because $L_x$ inverts the sign assigned by $S$ on each state, i.e., $L_x S L_x^{-1} = -S$ (see Fig. 1b).

If more than one translation operators anti-commute with $S$, one can always recombine these translation operators $L_i$ to form translation operators $L_i'$, where only one of $L_i'$ anti-commutes with $S$. Importantly, the two sets of translation operators, $L_i$ and $L_i'$, correspond to the same primitive unit cells. For instance, if $\{L_x, S\} = \{L_y, S\} = 0$, we can make the recombination $L_d = L_x L_y$ with $[L_d, S] = 0$. Then, $L_x$ and $L_d$ form another set of translation operators for the same primitive unit cells.

Thus, without loss of generality, to analyze the bulk topology, we assume $L_x$ as the only translation operator that anti-commutes with $S$ hereafter.

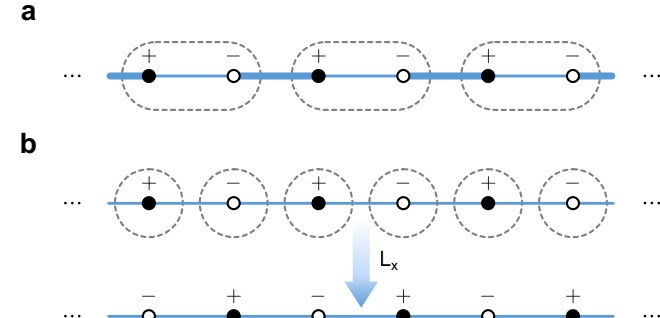

**Fig. 1 | Models for two classes of sublattice symmetries.** $A$ and $B$ sublattice sites are denoted by solid and hollow circles, respectively, and the signs $\pm$ at lattice sites are assigned by the sublattice symmetry. The primitive unit cells are indicated by the dashed loops. **a** The SSH model. The sublattice labeling and the unit cells have the same periodicity. **b** The 1D lattice with uniform nearest neighbor hopping amplitudes. The period of unit cells is a half of that of the sublattice labeling. The unit-cell translation $L_x$ exchanges two sublattices and therefore inverts $\pm$ signs.

### Two representations of symmetry algebra

To analyze the implications of Eq. (3) on band structures, we introduce two natural conventions to define the $k$ space, primitive unit cells and double unit cells, which correspond to two equivalent representations of the symmetry algebra in Eq. (3).

In the first convention, the $k$ space is defined from the primitive unit cells, i.e., the primitive translation operator $L_x$ is represented by $L_x = e^{ik_x a}$ with $a$ the lattice constant. As Eq. (3) can be written as $S L_x S^{-1} = -L_x$, we obtain $S e^{ika} S^{-1} = -e^{ika} = e^{i(k+\pi/a)a}$. This leads to a significant consequence: $S$ translates $k_x$ to $k_x + \pi/a$ with $\pi/a$ a half reciprocal lattice constant. Hereafter, we set $a = 1$ for simplicity, and accordingly

$$S : k_x \mapsto k_x + \pi. \tag{4}$$

Let $U_S$ be the $k$-space unitary operator of $S$. The symmetry identity $\{H, S\} = 0$ is represented in the $k$ space by

$$U_S \mathcal{H}^{(p)}(k_x, \bar{k}) U_S^\dagger = -\mathcal{H}^{(p)}(k_x + \pi, \bar{k}), \tag{5}$$

where $\bar{k}$ denotes the remaining components of $k$ except for $k_x$. One may formally write $S = U_S \mathcal{L}_\pi^x$ in $k$ space with $\mathcal{L}_\pi^x$ the translation operator of $k_x$ by $\pi$. Then, $\{H, S\} = 0$ is manifestly equivalent to Eq. (5).

The consequence of Eq. (5) in band structure is that for each eigenstate $|u(k_x, \bar{k})\rangle$ with energy $\mathcal{E}(\boldsymbol{k})$, the transformed state $U_S|u(k_x, \bar{k})\rangle$ satisfies

$$\mathcal{H}^{(p)}(k_x + \pi, \bar{k}) U_S|u(k_x, \bar{k})\rangle = -\mathcal{E}(\boldsymbol{k}) U_S|u(k_x, \bar{k})\rangle. \tag{6}$$

Hence, the band structure represented in the $(\mathcal{E}, \boldsymbol{k})$ space features a glide reflection symmetry, i.e., it is invariant under the coordinate transformation,

$$(\mathcal{E}, k_x, \bar{k}) \mapsto (-\mathcal{E}, k_x + \pi, \bar{k}). \tag{7}$$

The energy–momentum glide reflection symmetry is demonstrated by the single band illustrated in Fig. 2a.

In the second convention, we double the unit cells, i.e., each unit cell consists of two neighboring primitive unit cells along the $x$ direction. The translation operator that transforms each doubled unit cell to its nearest neighbor is the square $L_x^2$, and therefore the $k_x$ coordinate is defined by $L_x^2 = e^{ik_x \cdot 2a}$. To simplify the notation, we set $2a = 1$. Since

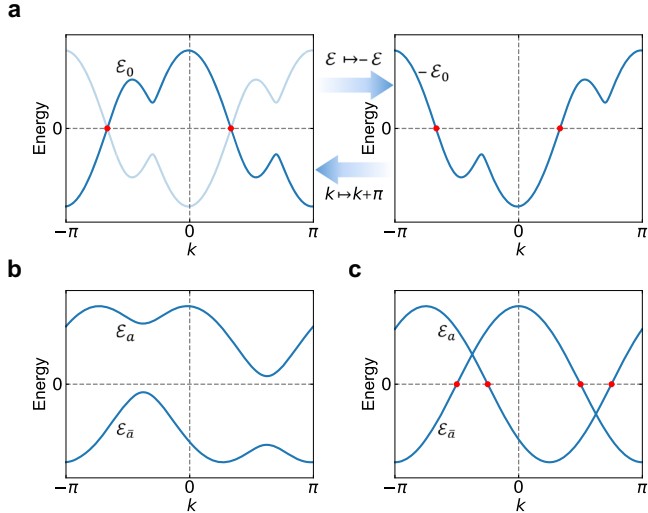

**a**

**b**

**c**

**Fig. 2 | Two elementary scenarios for zero modes. a** A single band preserving the glide reflection in the $(\mathcal{E}, k)$ space with two zero modes. The band is plotted in the left panel. Its energy-inverted image is plotted in the right panel and marked in light blue in the left panel for reference. The energy-inverted image is related to the original band by $k \mapsto k + \pi$. **b** A pair of gapped bands preserving the glide reflection. **c** A pair of bands preserving the glide reflection with four zero modes.

$[S, L_x^2] = 0$, $S$ can be represented in the usual way as

$$U_S = \tau_z \otimes 1_N, \tag{8}$$

where $N$ is the number of states in a primitive unit cell and $\tau$'s are the Pauli matrices acting in the sublattice space. As usual, $\{\mathcal{H}^{(d)}(\boldsymbol{k}), U_S\} = 0$ leads to

$$\mathcal{H}^{(d)}(\boldsymbol{k}) = \begin{bmatrix} 0 & Q(\boldsymbol{k}) \\ Q^\dagger(\boldsymbol{k}) & 0 \end{bmatrix}_\tau, \tag{9}$$

with $Q(k)$ an $N \times N$ matrix.

It is significant to note that for the doubled unit cells the primitive translation operator $L_x$ acts as a nontrivial unitary operator in the $k$ space[36], and takes the general form,

$$\mathcal{L}_x^{(d)} = \begin{bmatrix} 0 & R(k_x) \\ e^{ik_x} R^\dagger(k_x) & 0 \end{bmatrix}_\tau, \tag{10}$$

where $R(k_x)$ is a unitary operator and the concrete expression can be found in Supplementary Note 1. The commutation relation $[\mathcal{L}_x^{(d)}, \mathcal{H}(\boldsymbol{k})] = 0$ leads to

$$Q^\dagger(\boldsymbol{k}) = e^{ik_x} R^\dagger(k_x) Q(\boldsymbol{k}) R^\dagger(k_x). \tag{11}$$

Then the Hamiltonian can be transformed to be $U\mathcal{H}^{(d)}(\boldsymbol{k})U^\dagger = \tau_x \otimes h(\boldsymbol{k})$ with $h(\boldsymbol{k}) = e^{ik_x/2} R^\dagger(k_x) Q(\boldsymbol{k})$ being Hermitian. More details are given in the section of reduced Hamiltonian in the doubled unit cell convention in Methods.

Two conventions correspond to two equivalent representations of the symmetry algebra in Eq. (3). The energy band structure of $\mathcal{H}^{(d)}(\boldsymbol{k})$ can be obtained from folding that of $\mathcal{H}^{(p)}(\boldsymbol{k})$. A general description of band folding can be found in ref.[36]. Below, let us proceed to derive the physical consequences for both gapless and gapped cases using the two representations, respectively.

## Zero-mode numbers

For the implications of class-II sublattice symmetry on metals or semimetals, it is sufficient to consider 1D systems without loss of

generality, and it is convenient to work under the convention with primitive unit cells. Here, we adopt the terminology "zero mode", which refers to a crossing point of the energy bands at zero energy. For $d$D systems, a zero mode will extend into a $(d-1)$D Fermi surface across the Brillouin zone.

The symmetry algebra in Eq. (3) can lead to strong constraints on zero modes because of the resultant glide reflection symmetry in band structure [see Eqs. (6) and (7)]. There are two elementary scenarios to consider.

In the first, a single band $\mathcal{E}_0(k)$ preserves the glide reflection symmetry, i.e.,

$$\mathcal{E}_0(k + \pi) = -\mathcal{E}_0(k). \tag{12}$$

If $\mathcal{E}_0(k_0) > 0$, then $\mathcal{E}_0(k_0 + \pi) = -\mathcal{E}_0(k_0) < 0$. Then, generically there are $2n + 1$ zero modes in the interval $[k_0, k_0 + \pi)$ with $n = 0, 1, 2, \cdots$. Furthermore, the energy curve in the interval $[k_0 + \pi, k_0 + 2\pi)$ is determined by that in $[k_0, k_0 + \pi)$ through the glide reflection symmetry (see Fig. 2a). Consequently, in each $2\pi$ period, generically there exist $4n + 2$ zero modes.

In the second, a pair of bands $\mathcal{E}_a(k)$ and $\mathcal{E}_{\bar{a}}(k)$ together preserve the glide reflection symmetry, i.e.,

$$\mathcal{E}_a(k + \pi) = -\mathcal{E}_{\bar{a}}(k). \tag{13}$$

If the two bands do not cross, then the band structure has a gap and there are no zero modes (see Fig. 2b). But, if the two bands have one crossing point at $k_0 \in [0, \pi)$, there exists another crossing point at $k_0 + \pi \in [\pi, 2\pi)$ because of Eq. (13) (see Fig. 2c). As each crossing point corresponds to two zero modes, the number of zero modes is a multiple of four, namely $4n$.

From the two elementary scenarios, we can conclude the following results for the aforementioned two cases of class-II sublattice symmetry. i) Each primitive unit cell contains an odd number of states. Since there exist odd unpaired single bands, the number of zero modes is equal to $4n + 2$. ii) Each primitive unit cell contains an even number of states. Since there exist even unpaired single bands, the number of zero modes is equal to $4n$.

This can be explicitly verified for two 1D lattice models in Fig. 3a, b. In Fig. 3c, d, we plot the energy spectrum of two models. For the model in Fig. 3a, each unit cell consists of three sites, and therefore, it should have $4n + 2$ zero modes according to our theory. This is verified by the band structure plotted in Fig. 3c, where we observe six zero modes, namely, $n = 1$. For the model in Fig. 3b, the unit cell contains two sites. Indeed, we observe eight zero modes in the band structure in Fig. 3d, consistent with the symmetry constraint of $4n$ zero modes.

Moreover, the constraints on zero modes can be applied to understand zero modes in the famous Hofstadter model with $\Phi = 2\pi p/q$ flux per plaquette[37,38]. Here, $p$ and $q$ are coprime integers. The two cases of $q$ being even and odd correspond precisely to the presence of even and odd sites in each unit cell, respectively. Hence, odd (even) $q$ leads to $4n + 2$ ($4n$) zero modes. More details about the Hofstadter model are given in the Supplementary Note 2.

## Topological classification and invariants for insulators

Let us proceed to consider the topological classification of the insulators with class-II sublattice symmetry. As previously emphasized, an insulator exists only if each unit cell contains even states.

To derive the topological classifications, it is more convenient to work with the doubled unit cell convention. In this convention, the symmetry algebra can be encapsulated as

$$\{\mathcal{L}^{(d)}, S\} = 0, \quad [\mathcal{L}^{(d)}]^2 = e^{ik_x}, \quad S^2 = 1,$$
$$\{S, \mathcal{H}^{(d)}(\boldsymbol{k})\} = 0, \quad [\mathcal{L}^{(d)}, \mathcal{H}^{(d)}(\boldsymbol{k})] = 0. \tag{14}$$

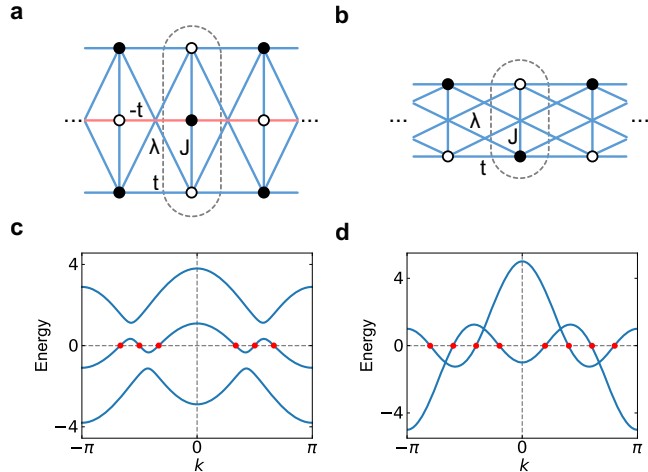

**Fig. 3 | Two lattice models of semimetals. a** A chain with three sites in each unit cell and a flux of $\pi$ threading each plaquette. **b** A chain with two sites in each unit cell and without flux. $t$ and $J$ stand for the nearest-neighbor hopping amplitudes along the $x$ and $y$ directions, respectively. $\lambda$ denotes the long-range hopping amplitude. All hopping amplitudes are real, and positive and negative ones are marked in blue and red, respectively. The primitive unit cells are surrounded by the dashed loops. **c, d** plot the band structures of the models with $t = J = \lambda = 1.0$ in (**a**) and (**b**), respectively. There are six and eight zero modes in (**c**) and (**d**), respectively.

**Table 1 | Topological classification table of insulators for two classes of sublattice symmetries**

| Class | Alg | Dim | 1 | 2 | 3 | 4 | 5 | 6 | 7 | 8 |
|-------|-----|-----|---|---|---|---|---|---|---|---|
| I | $[S, L_x] = 0$ | | $\mathbb{Z}$ | 0 | $\mathbb{Z}$ | 0 | $\mathbb{Z}$ | 0 | $\mathbb{Z}$ | 0 |
| II | $\{S, L_x\} = 0$ | | 0 | $\mathbb{Z}_2$ | 0 | $\mathbb{Z}_2$ | 0 | $\mathbb{Z}_2$ | 0 | $\mathbb{Z}_2$ |

"Dim" and "Alg" stand for spatial dimensionality and algebraic relation, respectively. The class-I $S$ is equivalent to the chiral symmetry in class AIII in the tenfold Altland–Zirnbauer symmetry classes.

Based on the algebraic relations, the derivation of the topological classification table, namely Table 1, has been provided in the section of topological classifications in Methods.

Compared to the topological classifications for class AIII or class-I sublattice symmetry, we observe from Table 1 two significant differences: i) The classification for class-II sublattice symmetry is nontrivial in even dimensions, while it is nontrivial in odd dimensions for class AIII; ii) The nontrivial classification for class-II sublattice symmetry corresponds to $\mathbb{Z}_2$ rather than $\mathbb{Z}$ for class AIII.

To understand why the classification is trivial in odd dimensions for class-II sublattice symmetry, it is noteworthy that under the doubled unit cell convention the winding numbers of $Q(k)$ in Eq. (9) vanish for all odd dimensions as a consequence of the algebraic relations (14). The derivation details are provided in the section of vanishing winding numbers in Methods.

The $\mathbb{Z}_2$ topological invariants in even dimensions originate from the symmetry constraint (5), namely the glide reflection symmetry in the $(\mathcal{E}, \boldsymbol{k})$ space. The topological invariants can be formulated under both conventions. Here, we demonstrate the formulation under the primitive unit cell convention. The essential idea of formulating these topological invariants can be illustrated in two dimensions.

For $\mathcal{H}^{(p)}(k_x, k_y)$, let us consider the Berry phases $\gamma_\pm^y(k_x)$ of conduction and valence bands for a 1D $k_y$-subsystem $\mathcal{H}^{1D}_{k_x}(k_y) := \mathcal{H}^{(p)}(k_x, k_y)$ with given $k_x$. Here, the Berry phases are defined as $\gamma_\pm^y(k_x) = \oint dk_y \, a_\pm^y(k_x, k_y)$ with $a_\pm^\mu := i\mathrm{Tr}\mathcal{A}_\pm^\mu$, and the Berry connections $\mathcal{A}_\pm^\mu$ are defined as $[\mathcal{A}_\pm^\mu]_{ab} = \langle \pm, a, \boldsymbol{k}| \partial^\mu | \pm, b, \boldsymbol{k} \rangle$ from the conduction and valence states $|\pm, a, \boldsymbol{k}\rangle$, respectively.

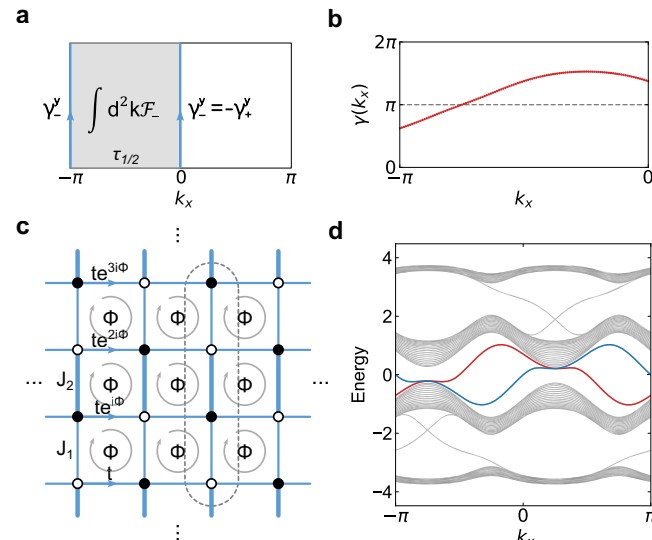

**Fig. 4 | Topological invariant and topological edge states. a** The Berry flux through one half of the BZ and the Zak phases of the 1D subsystems with $k_x = \pm \pi$ and 0. **b** The Berry phase $\gamma(k_x)$ as a function of $k_x$, which crosses $\pi$ once in one half of the BZ. **c** Schematic of the dimerized Hofstadter model with $\Phi = \pi/2$. $J_1$ and $J_2$ stand for the two alternatively distributed hopping amplitudes along the $y$ direction. $t$ denotes the hopping amplitude along the $x$ direction, and $\Phi$ denotes the magnetic flux per plaquette. **d** The band structure of the model in **c** with the slab geometry. The edge states at two boundaries are marked in blue and red, respectively. The parameter values are chosen as $t = J_1 = 1.0$ and $J_2 = 2.0$.

We consider the half Brillouin zone (BZ) $\tau_{1/2}$ with $k_x \in [-\pi, 0)$ as illustrated in Fig. 4a. For the boundary 1D systems $\mathcal{H}^{1D}_0$ and $\mathcal{H}^{1D}_{-\pi}$, we have $U_S \mathcal{H}^{1D}_0 U_S^\dagger = -\mathcal{H}^{1D}_{-\pi}$ from Eq. (5). Hence, the valence states of $\mathcal{H}^{1D}_{-\pi}$ and the conduction states of $\mathcal{H}^{1D}_0$ are related by the unitary transformation $U_S$, and therefore $\gamma_-^y(-\pi) = \gamma_+^y(0) \bmod 2\pi$. It is well known that the sum of $\gamma_\pm$ for a 1D insulator is equal to zero modulo $2\pi$, i.e., $\gamma_+^y(0) + \gamma_-^y(0) = 0 \bmod 2\pi$. Hence, we can get $\gamma_-^y(-\pi) + \gamma_-^y(0) = 0 \bmod 2\pi$. Further considering the general identity $\int_{\tau_{1/2}} d^2k f_-(k_x, k_y) + \gamma_-^y(-\pi) - \gamma_-^y(\pi) \in 2\pi\mathbb{Z}$ from Stokes' theorem[39], we can formulate the $\mathbb{Z}_2$ topological invariant as

$$\nu = \frac{1}{2\pi} \int_{\tau_{1/2}} d^2k f_-(k_x, k_y) + \frac{1}{\pi} \gamma_-^y(-\pi) \bmod 2. \tag{15}$$

Here, $f_-(k_x, k_y) := \partial^{k_x} a_-^y - \partial^{k_y} a_-^x$ is the Abelian Berry curvature of valence bands of $\mathcal{H}^{(p)}(k_x, k_y)$. The above reasoning has justified that the formula is valued in integers. In fact, its integer value is gauge invariant only modulo 2, since a gauge transformation transforms $\gamma_-^y(-\pi)$ to $\gamma_-^y(-\pi) + 2\pi n$ with $n$ the winding number of the gauge transformation. We note that topological invariants of analogous form appeared previously but with completely different symmetry origins for quantization[39,40].

If we can derive the Berry phase $\gamma_-^y(k_x)$ as a smooth function of $k_x$, the topological invariant is nontrivial if and only if $\gamma_-^y(k_x)$ crosses $\pi$ odd times[40–42]. Therefore, for an insulator with nontrivial $\nu = 1$, there must be in-gap edge states located at each edge parallel to the $x$ direction. A geometric interpretation of the $\mathbb{Z}_2$ topological invariant (15) and its implications to the bulk-boundary correspondence have been presented in the section of bulk-boundary correspondence in Methods.

The 2D topological insulators can be demonstrated by dimerized Hofstadter models. Specifically, we consider the dimerized Hofstadter model with $\Phi = \pi/2$ (see Fig. 4c). The topological invariant (15) can be read off from the flow of the Berry phase $\gamma_-^y(k_x)$ (see Fig. 4b). Since it crosses $\pi$ once in $k_x \in [-\pi, 0)$, the $\mathbb{Z}_2$ topological invariant (15) is nontrivial. The band structure of the model on a slab geometry with

the finite $y$ dimension is plotted in Fig. 4d. We observe a pair of topological edge states appearing on the two edges, respectively. Notably, the spectrum of each edge preserves the sublattice symmetry, since it is invariant under the energy–momentum glide reflection symmetry in Eq. (7). Additionally, the edge band may be detached from the bulk bands, depending on the parameter values and boundary termination of the model.

Alternatively, the $\mathbb{Z}_2$ topological invariants (15) can equally well be formulated in the doubled unit cell convention, where a reduced Hamiltonian $h(k)$ naturally arises with "twisted" boundary conditions $h(k_x + 2\pi, \bar{\boldsymbol{k}}) = -h(k_x, \bar{\boldsymbol{k}})$ (see the section of reduced Hamiltonian in the doubled unit cell convention in Methods). Instead of the half BZ, we just apply all the rationale on the whole BZ for $h(k)$.

The 2D topological invariant (15) can be readily generalized to any higher even dimensions $2n$ by replacing the Berry curvature and Berry phase by their higher dimensional counterparts, namely the $n$th Chern character and Chern–Simons form, respectively[43]. For more details, see the section of topological invariants in Methods.

## DISCUSSION

In conclusion, by revealing the intrinsic spatial nature of sublattice symmetry, we discovered the class-II sublattice symmetry. Unlike the class-I sublattice symmetry, the class-II sublattice symmetry cannot be regarded as chiral symmetry in the tenfold symmetry classes. It is represented as the glide reflection symmetry of the energy band structure, which reverses the energy and translates one momentum coordinate by half of the reciprocal lattice constant. As a result, it introduces novel constraints on zero modes and leads to a different topological classification table.

Class-I and class-II sublattice symmetries are distinguished by whether the spatial period of the hopping amplitudes is the same as that of the bipartition of sublattices. Just like class-I sublattice symmetry, class-II sublattice symmetry ubiquitously exists in crystals as long as the hopping within the same sublattice is ignorable. Therefore, our work has wide applicability for the analysis of crystalline condensed matter and the design of artificial crystals, considering the distinct properties of class-II sublattice symmetry. For instance, the dimerized Hofstadter models studied in our work are important models in condensed matter physics, but their symmetry structure revealed here has long been unrecognized. Various metamaterials have rapidly expanded with their high tunability to simulate crystalline topological phases. Our revealing of the spatial nature of sublattice symmetry provides a structuring principle for metamaterials and paves the way for realizing these novel topological phases.

Based on the fundamentals established here, one can further explore how the class-II sublattice symmetry can greatly enrich symmetry-protected topological phases. It can diversify symmetry classes by including time-reversal symmetry, particle–hole symmetry, and crystal symmetry. Therefore, it provides an extended framework for exploring topological phases, similar to what has been done with class-I sublattice symmetry. Moreover, since sublattice symmetry has played a significant role in the development of non-Hermitian topological physics[44–48], it is interesting to explore the implications of our theory for non-Hermitian topological phases.

## Methods
### Reduced Hamiltonian in the doubled unit cell convention
In this section, we show that the symmetry algebra in Eq. (14) leads to essentially the same symmetry constraint as Eq. (5).

Due to the translation symmetry in Eq. (10), the Hamiltonian can always be transformed to be

$$U\mathcal{H}^{(d)}(\boldsymbol{k})U^\dagger = \tau_x \otimes h(\boldsymbol{k}). \tag{16}$$

Here, $U(k_x) = \mathrm{diag}(e^{ik_x/2}R^\dagger(k_x), 1_N)$ and

$$h(\boldsymbol{k}) = e^{ik_x/2}R^\dagger(k_x)Q(\boldsymbol{k}), \tag{17}$$

where Eq. (11) has been used. It is significant to notice that $h(k)$ is Hermitian, $h^\dagger(k) = h(k)$, and hence can be regarded as a "reduced Hamiltonian". However, the reduced Hamiltonian $h(k)$ is not periodic for $k_x$, but satisfies the "twisted" boundary conditions

$$h(k_x + 2\pi, \bar{\boldsymbol{k}}) = -h(k_x, \bar{\boldsymbol{k}}). \tag{18}$$

Thus, comparing the two equations (18) and (5), we arrive at essentially the same symmetry constraint in both conventions: Translating the (reduced) Hamiltonian by a certain length along $k_x$ inverts it. It is this symmetry constraint that naturally gives rise to the $\mathbb{Z}_2$ topological invariants in even dimensions (see Table 1).

### Vanishing winding numbers
In one dimension, for the Hamiltonian in Eq. (9) with doubled unit cells, one might still try to calculate the winding number of class AIII following the conventional prescription

$$\nu = \frac{1}{2\pi i} \oint dk \, [D(k)]^{-1} \partial_k D(k), \tag{19}$$

with $D(k) = \det Q(k)$. It is straightforward to derive from Eq. (17) that $\det Q(k) = \det h(k) \det[e^{-ik/2}R(k)]$. $\det Q(k)$ is completely determined by the translation operator rather than the concrete form of the Hamiltonian, which already indicates that $Q(k)$ is topologically trivial. By using the concrete form of $R(k)$, we have $\det R(k) = e^{iMk}$ and therefore $\det Q(k) = \det h(k)$, which is a real-valued function since $h(k)$ is Hermitian and hence gives a zero winding number. Here, we have assumed that there are $M$ A-sites and $M$ B-sites in each primitive unit cell.

In higher odd dimensions $2n + 1$ with $n \geq 1$, the $\mathbb{Z}$-invariant in class AIII is given by

$$W[\tilde{Q}] = C_n \int_{BZ} \mathrm{tr}(\tilde{Q}d\tilde{Q}^\dagger)^{2n+1}, \tag{20}$$

with $C_n = -n!/[(2n+1)!(2\pi i)^{n+1}]$. Here, $\tilde{Q}$ is given by the usual definition of flattened Hamiltonian as $\tilde{Q}(\boldsymbol{k}) = R(k_x)\tilde{h}(\boldsymbol{k})e^{-ik_x/2}$ where $\tilde{h}$ is Hermitian unitary with $\tilde{h}^2 = 1$. Using the well-known identity $W[UV] = W[U] + W[V]$, we have

$$W[\tilde{Q}(\boldsymbol{k})] = W[R(k_x)] + W[\tilde{h}(\boldsymbol{k})e^{-ik_x/2}]. \tag{21}$$

The first term on the right-hand side vanishes because $R$ only depends on $k_x$. Recalling that $W[\tilde{h}]$ vanishes for $\tilde{h}$ is Hermitian unitary, the vanishing of $W[\tilde{h}(\boldsymbol{k})e^{-ik_x/2}]$ can be understood.

Under the convention with primitive unit cells, the Hamiltonian $\mathcal{H}^{(p)}$ cannot be converted into an anti-diagonal form, and therefore the definition of winding number for class AIII does not hold anymore. One might still intend to calculate the Zak phase $\gamma$ of valence bands, which turns out still be trivial.

With the doubled unit cells, the winding number $\nu$ is well defined as above and $\gamma = \pi\nu$ mod $2\pi$. The Zak phase is invariant under doubling the unit cells or folding the BZ. Since $\nu = 0$, $\gamma$ must be trivial. Alternatively, one can directly prove $\gamma = 0$ mod $2\pi$ with primitive unit cells.

When in the presence of energy gap at half-filling, among the $2M$ bands, there are $M$ conduction bands and $M$ valence bands, which form $M$ pairs according to the sublattice symmetry [see Eq. (6)]. Hence, we may label the $M$ pairs by $(n, -n)$ with $n = 1, 2, \cdots, M$. Then, Eq. (6) implies the following identity for the Berry connection of each energy

band

$$a_n(k+\pi) = a_{-n}(k), \tag{22}$$

with $a_n(k) = \langle u_n(k)|i\partial_k|u_n(k)\rangle$. The Berry phase of the half-filling gap is

$$\gamma = \sum_{n=1}^{M} \oint dk\, a_{-n}(k) = \sum_{n=-M}^{M} \int_0^\pi dk\, a_n(k), \tag{23}$$

which can be determined by

$$\gamma = i\ln\frac{\det[U(\pi)]}{\det[U(0)]}. \tag{24}$$

where $U(k) = (|u_{-M}(k)\rangle, \cdots, |u_{+M}(k)\rangle)$. Moreover, class-II sublattice symmetry ensures $\det[U(k+\pi)] = \det[U(k)]$, indicating a trivial Berry phase, $\gamma = 0$.

## Topological classifications

The topological classification table in Table 1 exhibits strong topological classifications in the sense of strong topology in the tenfold topological classifications. That is, the Brillouin torus $T^d$ is reduced to the Brillouin sphere $S^d$ as the base space of the topological classification. The standard spherical coordinates are $(\phi, \theta_1, \cdots, \theta_{d-1})$ with $\phi \in [0, 2\pi)$ and $\theta_i \in [0, \pi]$.

We now consider a general scenario, namely the topological classification of gapped Hamiltonians $\mathcal{H}^d(\boldsymbol{k})$ with $k \in S^d$ under the symmetry constraints,

$$\{\mathcal{H}^d(\boldsymbol{k}), \Gamma_i^d(\boldsymbol{k})\} = 0. \tag{25}$$

Here, $\{\Gamma_i^d(\boldsymbol{k}), \Gamma_j^d(\boldsymbol{k})\} = 2\delta_{ij}\lambda_i(\boldsymbol{k})$ with $\lambda_i(k)$ being functions from $S^d$ to $\mathbb{C}$, and $i = 1, 2, \cdots, M$ labels the set of symmetries.

The $d$D Hamiltonian can be mapped to the $(d+1)$D Hamiltonian,

$$\mathcal{H}^{d+1}(\boldsymbol{k}, \theta_d) = \sin\theta_d \tau_1 \otimes \mathcal{H}^d(\boldsymbol{k}) + \cos\theta_d \tau_2 \otimes 1 \tag{26}$$

with $\theta_d \in [0, \pi]$. Since $\mathcal{H}^{d+1}(\boldsymbol{k}, \theta_d)$ is constant at $\theta_d = 0$ and $\theta_d = \pi$, $\mathcal{H}^{d+1}$ is based on $S^{d+1}$. The $(d+1)$D Hamiltonian satisfies the symmetry constraints

$$\{\mathcal{H}^{d+1}(\boldsymbol{k}), \Gamma_\mu^{d+1}(\boldsymbol{k})\} = 0. \tag{27}$$

Here, $\mu = 0, 1, \cdots, M$, and

$$\Gamma_0^{d+1} = \tau_3 \otimes 1, \Gamma_i^{d+1} = \tau_1 \otimes \Gamma_i^d(\boldsymbol{k}). \tag{28}$$

For the previous $M$ symmetry operators, we still have $\{\Gamma_i^{d+1}(\boldsymbol{k}), \Gamma_j^{d+1}(\boldsymbol{k})\} = 2\delta_{ij}\lambda_i(\boldsymbol{k})$. More importantly, the emergent chiral symmetry $\Gamma_0^{d+1}$ is required, which anti-commutes with all $\Gamma_i^{d+1}$, namely $\{\Gamma_0^{d+1}, \Gamma_i^{d+1}(\boldsymbol{k})\} = 0$, and satisfies $(\Gamma_0^{d+1})^2 = 1$.

Furthermore, given any $(d+1)$D Hamiltonian $\mathcal{H}^{d+1}(\boldsymbol{k})$ with the above symmetry constraints, we can map it to the $(d+2)$D Hamiltonian,

$$\mathcal{H}^{d+2}(\boldsymbol{k}, \theta_{d+1}) = \sin\theta_{d+1}\mathcal{H}^{d+1}(\boldsymbol{k}) + \cos\theta_{d+1}\Gamma_0^{d+1}, \tag{29}$$

where $k \in S^{d+1}$. The Hamiltonian is based on $S^{d+2}$ because it is constant at $\theta_{d+1} = 0$ and $\pi$. Now, $\Gamma_0^{d+1}$ is broken, and the others are preserved with $\Gamma_i^{d+2} = \Gamma_i^{d+1}$. The symmetry algebra is restored to the case of $d$D Hamiltonian.

Notably, the two maps (26) and (29) are invertible for homotopy equivalence classes. Recall that two Hamiltonians are in the same homotopy class if and only if one can be deformed to the other with all symmetries preserved and the energy gap never closed. Thus, recursively applying the two maps leads to a sequence with the same

topological classifications. Moreover, all even (odd) dimensions correspond to the same symmetry algebra, and therefore are in the same symmetry class. This underlies the so-called twofold Bott periodicity for class A and AIII in the tenfold topological classifications, and also for the class-II sublattice symmetry in Table 1.

Let us return to a detailed elucidation of our problem. To fit the above construction, we recombine the symmetry operators as

$$\Gamma^d(k_x) = iS\mathcal{L}^{(d)}, S, \tag{30}$$

both of which anti-commute with the Hamiltonian. Here, the superscript "$d$" denotes the space dimensionality and "$(d)$" indicates the doubled unit cell convention. Below, we explicitly present the two maps for our classification problem.

$d = 1$ Let us start with considering the 1D Hamiltonian $\mathcal{H}^{1D}(k_x)$. It satisfies the symmetry constraint,

$$\{\mathcal{H}^{1D}(k_x), \Gamma^{1D}(k_x)\} = 0, \tag{31}$$

with $[\Gamma^{1D}(k_x)]^2 = e^{ik_x}$.

$d = 2$ Then, the 1D Hamiltonian $\mathcal{H}^{1D}(k_x)$ can be mapped to the 2D Hamiltonian,

$$\mathcal{H}^{2D}(k_x, \theta_1) = \sin\theta_1 \tau_1 \otimes \mathcal{H}^{1D}(k_x) + \cos\theta_1 \tau_2 \otimes 1 \tag{32}$$

under the symmetry constraints

$$\{\mathcal{H}^{2D}(k_x), \Gamma^{2D}(k_x)\} = 0, \{\mathcal{H}^{2D}(k_x), S\} = 0. \tag{33}$$

Here, $\Gamma^{2D}(k_x) = \tau_1 \otimes \Gamma^{1D}(k_x)$ and $S = \tau_3 \otimes 1$, with $[\Gamma^{1D}(k_x)]^2 = e^{ik_x}$ and $S^2 = 1$. Importantly, the two symmetry operators anti-commute with each other,

$$\{\Gamma^{2D}(k_x), S\} = 0. \tag{34}$$

$d = 3$ With the constant chiral symmetry operator $S$, the 2D Hamiltonian can be mapped to the 3D Hamiltonian,

$$\mathcal{H}^{3D}(k_x, \theta_1, \theta_2) = \sin\theta_2\mathcal{H}^{2D}(k_x, \theta_1) + \cos\theta_2 S. \tag{35}$$

With $\Gamma^{3D}(k_x) = \Gamma^{2D}(k_x)$, the symmetry constraint is given by

$$\{\mathcal{H}^{3D}(k_x), \Gamma^{3D}(k_x)\} = 0. \tag{36}$$

Recursively applying the two maps, we observe that all even dimensions correspond to class-II sublattice symmetry. The topological classifications are equal to that of $d = 1$. Without loss of generality, we assume the concrete symmetry operator,

$$\Gamma^{1D}(k_x) = \begin{bmatrix} 0 & 1_N \\ e^{ik_x}1_N & 0 \end{bmatrix}. \tag{37}$$

The flattened Hamiltonian is restricted to the general form by the symmetry,

$$\widetilde{\mathcal{H}}^{1D}(k_x) = \begin{bmatrix} A(k_x) & -ie^{-ik_x/2}B(k_x) \\ ie^{ik_x/2}B(k_x) & -A(k_x) \end{bmatrix}, \tag{38}$$

where $A$ and $B$ are Hermitian matrices with

$$A(k_x + 2\pi) = A(k_x), B(k_x + 2\pi) = -B(k_x). \tag{39}$$

Moreover, the condition $[\widetilde{\mathcal{H}}^{1D}(k_x)]^2 = 1$ is equivalent to

$$A^2 + B^2 = 1, [A, B] = 0, \tag{40}$$

which in turn just means the matrix $U(k_x) = A(k_x) + iB(k_x)$ is unitary. Then, the topological classification of the 1D system is equivalent to the classification of unitary-matrix valued functions under the twisted periodic condition,

$$U(k_x + 2\pi) = U^\dagger(k_x). \tag{41}$$

It is well known that such functions have a $\mathbb{Z}_2$ classification, corresponding to the parity of the winding number in a $4\pi$ period. Thus, class-II sublattice corresponds to the $\mathbb{Z}_2$ classification in even dimensions.

To see the trivial classification in odd dimensions, we can construct another sequence by the two maps. Then, the corresponding 1D system is just the Hamiltonian with an additional chiral symmetry $S = \tau_3 \otimes 1_N$ to the Hamiltonian (38). The consequence of $S$ simply corresponds to the elimination of $A$. Then, the Hamiltonian is fully characterized by the Hermitian unitary matrix distribution $B(k_x)$ with $B(k_x + 2\pi) = -B(k_x)$, which corresponds to trivial classification.

## Topological invariants

In this section, we present the general form of the $\mathbb{Z}_2$ topological invariants for class-II sublattice symmetry. Recall that the Berry connection of valence bands is defined as

$$[\mathcal{A}^\mu(\mathbf{k})]_{ab} = \langle -,a,\mathbf{k}|\partial^\mu| -,b,\mathbf{k}\rangle, \tag{42}$$

which gives the Berry curvature,

$$\mathcal{F}^{\mu\nu} = \partial^\mu \mathcal{A}^\nu - \partial^\nu \mathcal{A}^\mu + [\mathcal{A}^\mu, \mathcal{A}^\nu]. \tag{43}$$

To formulate the general form of topological invariants we introduce the differential forms of the Berry connection and curvature,

$$\mathcal{A} = \mathcal{A}^\mu dk_\mu, \mathcal{F} = \frac{1}{2}\mathcal{F}^{\mu\nu}dk_\mu \wedge dk_\nu. \tag{44}$$

Note that $\mathcal{F} = d\mathcal{A} + \mathcal{A}\mathcal{A}$, where the product of two forms is implicitly assumed as the wedge product. Then, the $n$th Chern character is given by

$$\mathrm{Ch}_n(\mathcal{F}) = \frac{1}{n!}\mathrm{Tr}\left(\frac{i\mathcal{F}}{2\pi}\right)^n \tag{45}$$

which is a $2n$ form. The Chern–Simons form is given by

$$Q_{2n-1}(\mathcal{A},\mathcal{F}) = \frac{1}{(n-1)!}\left(\frac{i}{2\pi}\right)^n \int_0^1 dt\, \mathrm{Tr}\,\mathcal{A}\mathcal{F}_t^{(n-1)} \tag{46}$$

where $\mathcal{F}_t = td\mathcal{A} + t^2\mathcal{A}\mathcal{A}$. Note that $Q_{2n-1}(\mathcal{A},\mathcal{F})$ is a $2n-1$ form[43]. Locally, the Chern character is the total derivative of the Chern–Simons form[43], i.e.,

$$\mathrm{Ch}_n(\mathcal{F}) = dQ_{2n-1}(\mathcal{A},\mathcal{F}) \tag{47}$$

Accordingly, the $\mathbb{Z}_2$ invariant in $2n$ dimensions is formulated as

$$\nu_{2n} = \int_{\tau_{1/2}} \mathrm{Ch}_n(\mathcal{F}) - 2\int_{\partial\tau_{1/2}^+} Q_{2n-1}(\mathcal{A},\mathcal{F})\, \mathrm{mod}\ 2. \tag{48}$$

Here, $\tau_{1/2}$ is the half BZ, namely $\tau_{1/2} = [-\pi, 0] \times T^{2n-1}$ and $\partial\tau_{1/2}^+$ is the boundary $T^{2n-1}$ with $k_x = -\pi$. Note that the two boundaries $\partial\tau_{1/2}^\pm$ have opposite Chern–Simons integrals modulo 2 as they are related by the class-II sublattice symmetry.

We note that the previously used identity $W[UV] = W[U] + W[V]$ can be understood as a consequence of the gauge transformation of the Chern–Simons integral. The Chern–Simons integral,

$$\mathrm{CS}_{2n+1}[\mathcal{A}] = \int_{T^{2n+1}} Q_{2n+1}(\mathcal{A},\mathcal{F}), \tag{49}$$

transforms as

$$\mathrm{CS}_{2n+1}(\mathcal{A}^U) - \mathrm{CS}_{2n+1}(\mathcal{A}) = W[U] \tag{50}$$

under the gauge transformation[43],

$$\mathcal{A}^U = U\mathcal{A}U^\dagger + UdU^\dagger. \tag{51}$$

Then, we consider the gauge transformation $UV$. It can be implemented directly as

$$\mathrm{CS}_{2n+1}(\mathcal{A}^{UV}) - \mathrm{CS}_{2n+1}(\mathcal{A}) = W[UV]. \tag{52}$$

Alternatively, we first implement the gauge transformation $V$ on $A^U$, which leads to

$$\mathrm{CS}_{2n+1}(\mathcal{A}^{UV}) - \mathrm{CS}_{2n+1}(\mathcal{A}^U) = W[V], \tag{53}$$

and then successively implement $U$ according to Eq. (50). Thus, we observe that $W[UV] = W[U] + W[V]$.

## Bulk-boundary correspondence

It is important to elucidate the bulk-boundary correspondence of topological invariants as has been done for the tenfold topological classifications[49]. Here, we present a geometric picture for the $\mathbb{Z}_2$ topological invariant (15) in two dimensions. This pump interpretation connects the bulk topological invariant to the edge states.

The BZ of $\mathcal{H}^{(p)}(\mathbf{k})$ can be partitioned into two parts, $\tau_{1/2} \cup \bar{\tau}_{1/2}$, with $\tau_{1/2} = [-\pi, 0) \times [-\pi, \pi)$. Due to the sublattice symmetry constraint (5), only one half of the BZ is independent. Specifically, knowing the band structure over $\tau_{1/2}$, we can map out the band structure over $\bar{\tau}_{1/2}$ by the sublattice symmetry. Therefore, it is sufficient to focus on $\tau_{1/2}$.

Since $k_y$ is periodic, we can write $\tau_{1/2} = [-\pi, 0) \times S^1$ as a cylinder. Over the fundamental domain $\tau_{1/2} = [-\pi, 0) \times S^1$, we can always choose a complete set of continuous states $|u_n(\mathbf{k})\rangle$, which are periodic along $k_y$. Then, the corresponding Abelian Berry connection $a_-^\mu(\mathbf{k})$ is also periodic along $k_y$. Accordingly, we can compute the Berry phase $\gamma_-^y(k_x)$ that is continuous from $k_x = -\pi$ to 0. From Stokes' theorem, we have

$$\int_{\tau_{1/2}} d^2k f_- \begin{aligned} &= \int_{-\pi}^0 dk_x \partial_{k_x}\gamma_-^y(k_x) \\ &= \gamma_-^y(0) - \gamma_-^y(-\pi). \end{aligned} \tag{54}$$

Hence, the $\mathbb{Z}_2$ topological invariant can be rewritten as

$$\nu = \frac{1}{2\pi}[\gamma_-^y(-\pi) + \gamma_-^y(0)]\, \mathrm{mod}\ 2. \tag{55}$$

The path of $\gamma_-^y(k_x)$ must cross 0 or $\pi$ when varying $k_x$ from $-\pi$ to 0. Then $\mathbb{Z}_2$ topological invariant can interpreted as

$$\nu = W_\pi\, \mathrm{mod}\ 2, \tag{56}$$

where $W_\pi$ denotes the number of times that $\gamma(k_x)$ crosses $\pi$. Therefore, we obtain a geometric interpretation of $\nu$, that is, $\nu$ is nontrivial if and only if $\gamma_-^y(k_x)$ crosses $\pi$ odd times (see Fig. 4b).

Hence, it is significant to observe that the nontrivial topological invariant $\nu = 1\, \mathrm{mod}\ 2$ ensures that there exists at least one $k_x^0 \in [-\pi, 0)$ with $\gamma_-^y(k_x^0) = \pi\, \mathrm{mod}\ 2\pi$. But, the quantized Berry phase $\pi$ of the 1D $k_y$-subsystem with $k_x = k_x^0$ implies the existence of an in-gap mode at each end. Hence, as reflected in the boundary BZ parametrized by $k_x$

together with the continuity of the band structure, the nontrivial topological invariant leads to a band that contains edge states in the band gap.

Two remarks are ready for the edge states. First, from the argument above, we see that the topological edge band does not necessarily connect valence and conduction bands in the bulk as in the case of the Chern insulator. It is also possible that the edge band is completely detached from the bulk bands. Second, the argument of above accounts for the topological edge band in the half of the BZ with $k_x \in [-\pi, 0)$. The topological edge band in the other half of the BZ with $k_x \in [-\pi, 0)$ can be mapped out from this by using the class-II sublattice symmetry. Moreover, by the continuity of band structure, generically the band for $k_x \in [-\pi, 0)$ crosses the zero energy, because $k_x = -\pi$ and $k_x = 0$ are related by the sublattice symmetry and therefore are of opposite energies.

Above we discussed the bulk-boundary correspondence for two dimensions. The arguments and generic features of the topological boundary states can be readily generalized to any even dimensions, where we use the pump of the Chern–Simons integral in a half of the BZ. Note that the specialization of the Chern–Simons integral in one dimension is just the Berry phase, and it is well-known that a half quantized Chern–Simons integral leads to boundary states.

## Detailed information for lattice models

We now present more information for lattice models that demonstrate our theory. Further details can be found in Supplementary Note 2.

For the gapless phase, we consider two 1D lattice models, as illustrated in Fig. 3a, b. For both models, the translation symmetry exchanges $A$ and $B$ sublattices, and therefore their sublattice symmetries fall into class II. In Fig. 3a, b, all hopping amplitudes are real, with positive and negative ones marked in blue and red, respectively. Each plaquette of the model in Fig. 3a carries $\pi$ flux. We set the values of parameters as $t = J = \lambda = 1.0$ in Fig. 3c, d.

For the topological insulator phase, we consider dimerized Hofstadter models, in which dimerization opens a gap at zero energy[50]. We choose to work in the special gauge configurations so that only the hoppings along $x$-direction pick up a nontrivial phase factor, as illustrated in Fig. 4c. We set the values of parameters as $\Phi = \pi/2$, $t = J_1 = 1.0$, and $J_2 = 2.0$ in Fig. 4b, d. The energy dispersion of these in-gap edge states can be easily obtained by the boundary effective theory, as demonstrated in Supplementary Note 3.

## Data availability

The data generated and analyzed during this study are available from the corresponding author upon request.

## Code availability

All code used to generate the plotted band structures can be found at https://doi.org/10.24433/CO.5326929.v1.

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

## Acknowledgements
This work is supported by the National Natural Science Foundation of China (Grants No. 12174181 and No. 12161160315), the Basic Research Program of Jiangsu Province (Grant No. BK20211506), and the startup fund of The University of Hong Kong.

## Author contributions
R.X. and Y.X.Z. conceived the idea. Y.X.Z. supervised the project. R.X. and Y.X.Z. did the theoretical analysis. R.X. and Y.X.Z. wrote the manuscript.

## Competing interests
The authors declare no competing interests.
