## [Peer Review File · Nature Communications]

Revealing the spatial nature of sublattice symmetryReviewers' Comments:

Reviewer #1:

Remarks to the Author:

In this work, the authors consider a subtle difference between the chiral symmetry (local symmetry obtained as composition of time reversal and charge conjugation) and sublattice symmetry (spatial symmetry that acts with opposite sign on two sublattices of a lattice).

In most of the existing body of literature on topological band theory, the two terms (i.e., chiral/sublattice symmetry) are used interchangeably, as both symmetries are represented with a unitary operator that anticommutes with the Hamiltonian: $\{H, S\} = 0$. However, here the authors reveal a more subtle relation that arises in the presence of translation symmetry. Namely, in that case two classes of sublattice symmetry can be distinguished, dubbed class-I vs. class-II, depending on whether the sublattice labelling is compatible with the primitive translations (equivalently: whether the number of sites per unit cell is even or odd). It is found that class-I sublattice symmetry is indeed mathematically equivalent to chiral symmetry; in contrast, class-II sublattice symmetry results in a distinct constraints on the Bloch Hamiltonian $H(k)$, which implies modification of the topological classification.

To my knowledge, this nuanced distinction between chiral and sublattice symmetry has not been explicitly discussed before. I find the authors manuscript to provide an interesting and valuable comment on the existing body of work on the characterization of topological bands. The methods seem appropriate, the presentation is mostly clear and structured, and the conclusions are sound. The work certainly deserves to be published in a specialized topical journal in some form.

However, when considering the suitability of the manuscript for the high-profile journal Nature Communication, one should also consider the broader impact expected by this work. In this regard, let me point out that the "sublattice realization" of chiral symmetry is not easily achieved in real materials, as it requires the unphysical tuning of intra-sublattice hopping amplitudes to zero. (This aspect is perhaps correlated with the fact that it took ~ 15 years for this subtle distinction between chiral vs sublattice symmetry to be noticed.) Therefore, the finding is of a rather academic interest, with only a limited experimental relevance for suitably tailored metamaterials.

The paper then discusses why class-II sublattice symmetry (1) trivializes the usual winding numbers in odd dimensions while (2) enabling a \mathbb{Z}_2 invariant in even dimensions. However, here I find that the discussion lacks the necessary rigor expected of a theoretical paper. On one hand, the mathematical steps leading from Eq.(14) to the classification result in Table I are not clarified; on the other hand, while bulk-boundary correspondence for the \mathbb{Z}_2 -invariant is numerically observed for a simple model, it is not proved or motivated from mathematical principles.

I further have the following questions and comments that the authors should consider in future versions of their work:

(1) For the \mathbb{Z}_2 -classified phase, could the nature of a topology-altering critical point be discussed? What is the effective theory in the bulk, and how do the edge states become trivialized?

(2) Could the authors provide more comments on the cases when the sublattices A and B have different number of lattice sites?

(3) When reading the manuscript, I was initially puzzled by the usage of the term "zero mode". To me, a "mode" suggest a localized or an otherwise isolated state in the spectrum. However, here the authors use it to mean a continuum bulk state at $E=0$. Furthermore, in spatial dimensions $d>1$, these "zero modes" in fact correspond to extended Fermi surfaces (so that the $4n$ vs $4n+2$ distinction corresponds to counting Fermi surfaces). It would help if the authors could include a proper

clarification of the terminology.

Reviewer #2:

Remarks to the Author:

Review of the manuscript NCOMMS-23-59837

'Revealing the spatial nature of sublattice symmetry'

by Rong Xiao and Y. X. Zhao

The authors find a seemingly overlooked piece of a tenfold classification of topological systems - a chiral invariant which unlike well-known winding number is non-vanishing only in even dimensions.

In my opinion the paper is clearly written and provides some striking new fundamental results. Therefore it probably deserves being published in Nature Communications journal. However, the authors should address a few remarks:

- 1) It is a bit misleading that around Eq. (14) one focuses on the Hamiltonian with doubled cell, which is not really used for anything, and then one uses a primitive-cell Hamiltonian for defining the invariant without any warning.
- 2) It should be explicitly written how one calculates the invariant (15) for a multiband system. It should be also noted that this is indeed the same invariant as the one studied in previous paper of this group, namely, Ref. 42 (even Fig. 4b is quite the same in both papers).
- 3) Why is spectrum of Fig. 4d not chiral-symmetric? Is the edge breaking chiral symmetry? Will the edge-states always connect the bands through the gap or it is termination-dependent?
- 4) How robust is bulk-boundary correspondence? It seems that one needs k -space to define the invariant. Then is the topological phase robust with respect to disorder that breaks translational invariance but keeps chiral symmetry?
- 5) For a standard chiral symmetry one can derive the bulk-boundary correspondence using Green's function formulation [e.g. PHYSICAL REVIEW B 84, 125132 (2011)]. Can it still be done in this case?
- 6) In Supplement above Eq. (12) it is said:
"Using the well-known identity $W[UV] = W[U] + W[V]$ "
I would like either a proper reference or derivation of this identity.

Referee comment: *In this work, the authors consider a subtle difference between the chiral symmetry (local symmetry obtained as composition of time reversal and charge conjugation) and sublattice symmetry (spatial symmetry that acts with opposite sign on two sublattices of a lattice).*

In most of the existing body of literature on topological band theory, the two terms (i.e., chiral/sublattice symmetry) are used interchangeably, as both symmetries are represented with a unitary operator that anticommutes with the Hamiltonian: $\{H, S\} = 0$. However, here the authors reveal a more subtle relation that arises in the presence of translation symmetry. Namely, in that case two classes of sublattice symmetry can be distinguished, dubbed class-I vs. class-II, depending on whether the sublattice labelling is compatible with the primitive translations (equivalently: whether the number of sites per unit cell is even or odd). It is found that class-I sublattice symmetry is indeed mathematically equivalent to chiral symmetry; in contrast, class-II sublattice symmetry results in distinct constraints on the Bloch Hamiltonian $H(k)$, which implies modification of the topological classification.

Our reply: We appreciate the Referee's insightful summary of our work, from which we are pleased to see that the main achievement of our study has been well understood by the referee.

For scientific communication purposes, we would like to share our perspective on the discovery of class-II sublattice symmetry with the referee.

The concept of chiral symmetry in condensed matter systems was initially introduced by Altland and Zirnbauer in the context of the Bogoliubov-de Gennes (BdG) Hamiltonian for superconductors. In this setting, chiral symmetry is a combination of time-reversal and particle-hole symmetries, making it an internal symmetry. When the notion was extended to non-interacting crystalline materials in the form of sublattice symmetry, it was by default considered as an internal symmetry, effectively ignoring the spatial nature of sublattice symmetry.

Our careful re-examination of the difference between superconductor systems and non-interacting crystalline materials in terms of symmetry representation led us to identify the previously overlooked subtle point. This re-examination was motivated by our recent systematic investigation of the projective representation of crystal symmetry.

From this case, we learn that each type of physical system possesses its unique aspects in symmetry representation. Therefore, additional attention may be required when transferring symmetry concepts from one system to another. This observation underscores the importance of our work in uncovering the subtle distinction between chiral and sublattice symmetries, which has broader implications for the study of crystalline materials and their topological properties.

Referee comment: *To my knowledge, this nuanced distinction between chiral and sublattice symmetry has not been explicitly discussed before. I find the authors manuscript to provide an interesting and valuable comment on the existing body of work on the characterization of topological bands. The methods seem appropriate, the presentation is mostly clear and structured, and the conclusions are sound. The work certainly deserves to be published in a specialized topical journal in some form.*

Our reply: As theoretical physicists, we sincerely appreciate the referee's positive comment that our work "*provides an interesting and valuable comment on the existing body of work on the characterization of topological bands,*" and we consider this as a very high evaluation.

We are also grateful to the referee for the referee's acknowledgement of the scientific quality of our work, stating that "*the methods seem appropriate, and the conclusions are sound.*" Furthermore, we appreciate the recognition of our manuscript's presentation as being "*mostly clear and structured.*"

This feedback not only encourages us in our research but also strengthens our belief that our work makes a significant contribution to the field of topological bands and has the potential to inspire further studies in this area.

Referee comment: *However, when considering the suitability of the manuscript for the high-profile journal Nature Communication, one should also consider the broader impact expected by this work. In this regard, let me point out that the "sublattice realization" of chiral symmetry is not easily achieved in real materials, as it requires the unphysical tuning of intra-sublattice hopping amplitudes to zero. (This aspect is perhaps correlated with the fact that it took ~15 years for this subtle distinction between chiral vs sublattice symmetry to be noticed.) Therefore, the finding is of a rather academic interest, with only a limited experimental relevance for suitably tailored metamaterials.*

Our reply: While we appreciate the referee's positive assessment of the scientific significance of our work, we respectfully disagree with the judgement that the expected impact of our work is not sufficient for Nature Communications.

The referee raised two specific points regarding this issue, and we would like to address them one by one below.

Specific point 1: The "sublattice realization" of chiral symmetry is not easily achieved in real materials, as it requires the unphysical tuning of intra-sublattice hopping amplitudes to zero.

Our response: We acknowledge that nonvanishing intra-sublattice hopping amplitudes typically exist in real materials, making any sublattice symmetry an approximate symmetry. However, this does not diminish the importance of sublattice symmetry as a topic with broad interests in condensed matter physics. Furthermore, this traditionally existing issue is neither new nor more severe in our case, as the models in our manuscript are natural, consisting of only nearest-neighbor hopping.

We would like to further note the following aspects:

1. Close to the Fermi surface, many real materials exhibit approximate sublattice symmetry with sufficient accuracy. Nontrivial physical consequences of sublattice symmetry are robust as long as the approximation accuracy is sufficient.
2. Several significant condensed-matter models possess sublattice symmetry, for instance, the Hofstadter models considered in our work. The importance of sublattice symmetry is justified by its ability to analyze structures in these essential models.

Specific point 2: The finding is of rather academic interest, with only limited experimental relevance for suitably tailored metamaterials.

Our response: The evaluation of general interest in metamaterials is subjective. Apart from the broad interests mentioned in condensed matter physics, we believe that our work's fundamental contribution to structuring metamaterials is already of significant broad interest. This is because various metamaterials, such as photonic/acoustic crystals, cold atoms in optical lattices, periodic mechanical systems, and even periodic electric arrays, have been a rapidly expanding field. These metamaterials have demonstrated numerous novel topological phases with their high tunability, which are otherwise difficult to achieve in real materials.

Furthermore, we would like to share our general perspective on crystal symmetry and metamaterials. From a mathematical physicist's viewpoint, the representation theory of crystal symmetry should be applicable to any form of crystals and is not limited to condensed matter systems like quantum materials, superconductors, and spin systems. Consequently, various metamaterials, including photonic/acoustic crystals, cold atoms in optical lattices, periodic mechanical systems, and even periodic electric arrays, can represent broader crystal symmetry classes with their high tunability, such as projective crystal symmetry.

Lastly, we appreciate the referee's note that "*it took ~15 years for this subtle distinction between chiral vs sublattice symmetry to be noticed.*" As mentioned earlier, this discovery arose from our systematic investigation of new crystal symmetry classes, making it an unusually exciting discovery for us.

In the revision of our manuscript, we have significantly extended the Discussion section to illuminate the broader impact of our work, addressing this issue raised by the referee. This revision also follows the editor's suggestion.

Referee comment: *The paper then discusses why class-II sublattice symmetry (1) trivializes the usual winding numbers in odd dimensions while (2) enabling a Z_2 invariant in even dimensions. However, here I find that the discussion lacks the necessary rigor expected of a theoretical paper. On one hand, the mathematical steps leading from Eq.(14) to the classification result in Table I are not clarified; on the other hand, while bulk-boundary correspondence for the Z_2 -invariant is numerically observed for a simple model, it is not proved or motivated from mathematical principles.*

Our reply: We would like to express our gratitude to the referee for pointing out two specific issues concerning the presentation of our results. While drafting the manuscript, our intention was to avoid overly technical content for the general readership of Nature Communications, which led us to exclude these detailed derivations. However, we appreciate the referee’s perspective on the necessity of including these technical details.

In response to the referee’s comments, we have incorporated three additional sections within the Methods section to address the two particular issues: 1. Topological Classifications, 2. Topological Invariants, and 3. Bulk-Boundary Correspondence.

Technical comment 1: *I further have the following questions and comments that the authors should consider in future versions of their work:*

(1) *For the Z2-classified phase, could the nature of a topology-altering critical point be discussed? What is the effective theory in the bulk, and how do the edge states become trivialized?*

Our reply: We are grateful to the referee for raising this issue.

The critical point in question is indeed represented by multiple Dirac points, which are analogous to the prior critical points observed in topological insulators. We kindly direct the referee to Fig. 1b and 1d, where the critical point of the Hofstadter model analyzed in our manuscript is depicted with a slab geometry and with full translational symmetry, respectively.

The trivialization of the edge states (refer to Fig. 1a) can be observed from Fig. 1b to Fig. 1c, with J_1 being increased. This figure illustrates the band structure of the Hofstadter model using a slab geometry, effectively demonstrating the trivialization.

Fig. 1. Energy spectrum of the dimerized Hofstadter model. **a**, **b** and **c** are depicted with a slab geometry, and **d** is depicted in the Brillouin zone with full translational symmetry. **a**. The topological nontrivial phase with $t = J_1 = 1.0$ and $J_2 = 2.0$. **b**. Critical phase with $t = J_1 = J_2 = 1.0$. **c**. Topological trivial phase with $t = J_2 = 1.0$ and $J_1 = 2.0$. **d**. The bulk band structure of the critical phase with $t = J_1 = J_2 = 1.0$. The Dirac points are marked by red dots. Here we set the parameter $\Phi = \pi/2$.

Technical comment 2: (2) *Could the authors provide more comments on the cases when the sublattices A and B have different number of lattice sites?*

Our reply: This is an excellent question.

The class-II sublattice symmetry discussed here still assumes an equal number of sublattice sites, similar to the class-I sublattice symmetry or chiral symmetry in the tenfold symmetry classes. The question raised by the referee challenges this assumption, and it is worth noting that sublattices with non-equal sublattices are indeed widespread, with examples such as the Lieb lattice and dice lattice. In this regard, classifying topological phases for these more general sublattice symmetries is a fundamental problem.

A characteristic of such sublattice symmetry is the presence of zero-energy flat bands, which are required by the symmetry. The existence of these flat bands further highlights the scientific importance of addressing this issue.

We would like to share with the referee that we have resolved the topological classifications for these sublattice symmetries, and the manuscript has been completed and is now prepared for submission. The current work, in conjunction with the recently completed work, together form a series of studies that comprehensively address sublattice symmetries for all bipartite lattices.

Technical comment 3: *(3) When reading the manuscript, I was initially puzzled by the usage of the term “zero mode”. To me, a “mode” suggests a localized or an otherwise isolated state in the spectrum. However, here the authors use it to mean a continuum bulk state at $E=0$. Furthermore, in spatial dimensions $d>1$, these “zero modes” in fact correspond to extended Fermi surfaces (so that the $4n$ vs $4n+2$ distinction corresponds to counting Fermi surfaces). It would help if the authors could include a proper clarification of the terminology.*

Our reply: We are grateful to the referee for pointing out the potential confusion arising from our usage of terminology. In the revised manuscript, we have taken the referee’s suggestion into account by providing a clear explanation of the terms used. We sincerely appreciate the referee’s meticulous review of our manuscript.

-----Response to Report of Referee #2 -----

Referee's overall comment: *The authors find a seemingly overlooked piece of a tenfold classification of topological systems - a chiral invariant which unlike well-known winding number is non-vanishing only in even dimensions.*

In my opinion the paper is clearly written and provides some striking new fundamental results. Therefore it probably deserves being published in Nature Communications journal.

Our reply: We are grateful for the referee's positive evaluation of our work, particularly for recognizing that our paper "*provides some striking new fundamental results.*" We are honored by the referee's assessment that our work merits publication in Nature Communications based on these findings.

We also value the referee's accurate summary of our work, highlighting it as an "*overlooked piece of a tenfold classification of topological systems.*" In response to this point, we wish to offer a potential explanation for why this aspect has been overlooked for so long. Specifically, the chiral symmetry within the tenfold symmetry classes was initially conceptualized as a combination of time-reversal and particle-hole symmetries within the framework of BdG Hamiltonians for superconductors, categorizing it as an internal symmetry. This concept was then directly applied to non-interacting crystalline systems. However, we argue that the sublattice symmetry, being fundamentally of spatial origin, should not be straightforwardly classified as an internal symmetry.

Additionally, we are thankful for the referee's recognition of the clarity of our paper's presentation.

Referee's specific comment: 1) *However, the authors should address a few remarks:*

1) It is a bit misleading that around Eq. (14) one focuses on the Hamiltonian with doubled cell, which is not really used for anything, and then one uses a primitive-cell Hamiltonian for defining the invariant without any warning.

Our reply: We appreciate the referee's insightful comments regarding the presentation of our work and agree that providing more explanations and comments will improve the logical flow.

Upon re-examining the relevant sections, we have identified two reasons for including the doubled cell representation of the sublattice symmetry, which we would like to share with the referee:

1) We have found that some readers may mistakenly believe that doubling the primitive unit cell can transform class-II sublattice symmetry into class-I. In fact, during discussions with our colleagues, we noticed that such confusion had arisen. To address this, we have explicitly discussed the doubled cell representation and demonstrated that the symmetry

algebra can remove the previous topological invariants (i.e., winding numbers) in odd dimensions. By doing so, we can prevent any potential confusion.

2) The primitive unit cell convention and the doubled unit cell convention offer two equivalent representations of the same symmetry algebra. Although the primitive unit cell convention is more natural, the doubled unit cell convention proves to be technically more convenient for deriving the topological classification. We have now included this derivation in a section within the Methods.

In the revised manuscript, we have incorporated these explanations and comments in the appropriate sections where the doubled unit cell convention is discussed. We believe these revisions will help clarify our work for the reader.

Referee's specific comment: 2) *It should be explicitly written how one calculates the invariant (15) for a multiband system. It should be also noted that this is indeed the same invariant as the one studied in previous paper of this group, namely, Ref. 42 (even Fig. 4b is quite the same in both papers).*

Our reply: We appreciate the referee's excellent suggestion regarding the presentation of the topological invariant, as the multiband case is indeed more general. In multiband systems, one can simply use the conventional method of defining the Abelian Berry connection as the trace of the non-Abelian Berry connection. Consequently, for numerical computation, the determinant of the Wilson loop operator can be used as the exponential function of the Abelian Berry connection.

As the referee has pointed out, the topological invariant in our work is indeed the same as the one examined in our previous paper. However, we would like to emphasize that although the expression is identical, the underlying reasons justifying this expression as the topological invariant are quite different between the two papers. In Ref. 42, the two 1D subsystems exhibit opposite Berry phases due to their relation by an inversion symmetry. In contrast, in our current study, the class-II sublattice symmetry connects the valence bands of one 1D system to the conduction bands of another 1D system, resulting in opposite Berry phases for the valence bands of the two 1D systems. In fact, topological invariants in this form, specifically flux subtracted by boundary Berry phases, first appeared in Fu and Kane's paper [PRB 74, 195312 (2006)] within the physics literature.

In the revised manuscript, we have updated the formulation of the topological invariant to accommodate multiband systems. Additionally, we have included a section on Topological Invariants in the Methods, which covers the formulation of topological invariants for all even dimensions. We have also highlighted that the topological invariant shares the same form as that in Ref. 42, but the underlying reasons are entirely different.

Referee's specific comment : 3) *Why is spectrum of Fig. 4d not chiral-symmetric? Is the edge breaking chiral symmetry? Will the edge-states always connect the bands through the gap or it is termination-dependent?*

Our reply: We thank the referee for raising the three questions.

For the first two questions. We would like to point out that in fact the band structure in Fig. 4d is chiral symmetric. It is a feature of the class-II sublattice symmetry that in the primitive unit cell convention the sublattice symmetry not only inverts the energy but also translates a momentum coordinate by π , i.e.,

$$E_a(k_x, k_y) \mapsto -E_{\bar{a}}(k_x, k_y + \pi),$$

because of the symmetry constraint on the Hamiltonian,

$$UH(k_x, k_y)U^\dagger = -H(k_x, k_y + \pi).$$

This operation on energy spectrum is referred to as the energy-momentum glide reflection symmetry of the band structure. We assume that the translational symmetry that anti-commutes with the sublattice symmetry is preserved in the slab geometry. Hence, the class-II sublattice symmetry is still reflected as the energy-momentum glide reflection symmetry. As shown in Fig. 2a, the energy-momentum glide reflection symmetry is preserved by the band structure.

Fig. 2. The band structures of the dimerized Hofstadter model in slab geometry under two conventions. **a.** The band structure in the primitive unit-cell convention. **b.** The band structure in the doubled unit-cell convention. The spectrum in **a** preserves the energy-momentum glide reflection symmetry, while the spectrum in **b** is symmetric with respect to zero energy. The parameter values are $t = J_1 = 1.0$, $J_2 = 4.0$, and $\Phi = \pi/2$.

Additionally, we would like to note that in the doubled unit-cell convention, the reflection of the class-II sublattice symmetry in the band structure is the same as that of the ordinary sublattice symmetry. For the referee's reference, we also present the band structure in Fig. 2b under this convention.

Then, let us proceed to the third question. In our case, the topological invariant does not require that the edge mid-gap bands connect bulk valence and conduction bands. Thus, the edge bands can be detached from the bulk bands, and whether the bands are connected through the gap is termination-dependent, as exemplified in Fig. 2a.

In the revised manuscript, we have added appropriate comments for these issues raised by the referee.

Referee's specific comment: 4) *How robust is bulk-boundary correspondence? It seems that one needs k -space to define the invariant. Then is the topological phase robust with respect to disorder that breaks translational invariance but keeps chiral symmetry?*

Our reply: We appreciate this issue raised by the referee, which highlights the fundamental difference between the spatial nature of class-II sublattice symmetry and chiral symmetry in the tenfold symmetry classes.

In the context of tenfold classifications, topological invariants can be expressed in real space. This allows for the invariants to be determined without strict translational symmetry, as they only necessitate the locality of real-space Hamiltonians. Essentially, the symmetries in these classifications are internal symmetries.

However, the situation differs when considering crystal symmetries. In this case, translational symmetry is assumed to be part of the symmetry group, and the topological invariants are formulated in k space. Consequently, a common challenge for most crystalline topological insulators is determining the robustness of boundary states in the presence of disorders that disrupt translational invariance. The general understanding is that boundary states should maintain their stability under weak disorders, as these states are energetically well-separated from the bulk states. We anticipate that the robustness of boundary states against weak disorders still holds for our topological phases.

This is verified by the dimerized Hofstadter model in Fig. 3. In this model, random hopping amplitudes are added between the A and B sublattices to maintain sublattice symmetry while disrupting translational invariance. Figure **a** displays the energy spectrum in relation to the disorder strength. The disorder strength w/J_1 , expressed in units of J_1 , signifies the relative scale of disorder strength compared to the model's hopping magnitudes. The zero-energy edge states can be seen to remain energetically separated from the bulk states even for significant disorders with $w/J_1 \sim 5.0$. However, as the disorder strength increases, the bulk spectrum broadens, eventually obscuring edge states at sufficiently high disorder strengths.

This finding is complementarily supported by Fig. **b**, which shows the average zero-energy local density of states for lattice sites at the top and bottom rows of the model as a function of the disorder strength w/J_1 . Figures **c-g** depict the spatial distributions of the zero-energy density of states corresponding to typical values of w/J_1 .

Fig. 3. The effects of disorders on in-gap edge states in the dimerized Hofstadter model. The hopping disorders depend on sites as $J_{A,B} \mapsto J_{A,B}^{i,j} = J_{1,2} + R_{i,j}$. Here, i (j) is the site index along x (y) direction and $R_{i,j}$ is a random variable with a uniform distribution over $[-w/2, w/2]$ with $\langle R_{i,j} \rangle = 0$. Such a hopping disorder breaks translational symmetry but preserves sublattice symmetry. **a.** The energy spectrum as a function of the disorder strength w/J_1 . The edge states can be clearly identified in the energy gap up to strengths over $w/J_1 \sim 5.0$. **b.** The average local density of states (LDOS) at zero-energy at the top and bottom rows as a function of the disorder strength w/J_1 . The distribution of LDOS over lattice sites for disorder strength **c.** $w/J_1 = 0.0$, **d.** $w/J_1 = 2.0$, **e.** $w/J_1 = 4.0$, **f.** $w/J_1 = 6.0$, and **g.** $w/J_1 = 8.0$. We set the parameter values as $t = J_1 = 1.0$, $J_2 = 4.0$, and $\Phi = \pi/2$. We choose the system size as $N_x \times N_y = 20 \times 20$ lattice sites and average over 100 disorder configurations to obtain the results.

Referee's specific comment: 5) *For a standard chiral symmetry one can derive the bulk-boundary correspondence using Green's function formulation [e.g. PHYSICAL REVIEW B 84, 125132 (2011)]. Can it still be done in this case?*

Our reply: It is indeed intriguing to consider whether the bulk-boundary correspondence can be derived from the Green's function formulation. We appreciate the referee's suggestion and reference to the paper [PRB 84, 125132 (2011)], which was one of the first papers read by the last author on the bulk-boundary correspondence of topological insulators and superconductors.

At this stage, we believe that it is possible, although we have not yet determined the exact method for doing so. The rationale behind this is that the nontrivial topological invariant formulated in our paper guarantees the existence of a sub-1D system with a Zak phase of π , which results in an edge mid-gap mode. Consequently, the nontrivial topological invariant leads to a mid-gap edge band. Since the bulk-boundary correspondence of a 1D system with a Zak phase of π can be derived using the Green's function formulation, it is reasonable to expect that the bulk-boundary correspondence in our case could also be derived using a Green's function formulation in some manner.

In the revised manuscript, we have cited [PRB 84, 125132 (2011)] along with an appropriate comment.

Referee's specific comment: 6) *In Supplement above Eq. (12) it is said: "Using the well-known identity $W[UV] = W[U] + W[V]$ " I would like either a proper reference or derivation of this identity.*

Our reply: We are grateful to the referee for raising this issue.

This identity is related to the gauge transformation of the Chern-Simons term. Let $SC_{2n+1}^k(A)$ be the $(2n+1)$ D Chern-Simons at level k . We adopt the normalization such that $e^{2\pi i SC_{2n+1}^k(A)}$ is well defined. Then, under a gauge transformation U , the Chern-Simons term is transformed as

$$SC_{2n+1}^k(A^U) = SC_{2n+1}^k(A) + kW(U).$$

A detailed derivation of this result can be found in Nakahara's book (Geometry, Topology and Physics).

Then, let us consider two successive gauge transformations U, V . There are two equivalent ways to the total gauge transformation, namely $SC_{2n+1}^k(A^{UV}) = SC_{2n+1}^k((A^U)^V)$. The two sides of the identity are calculated, respectively, below.

$$SC_{2n+1}^k(A^{UV}) = SC_{2n+1}^k(A) + kW(UV)$$

$$SC_{2n+1}^k((A^U)^V) = SC_{2n+1}^k(A^U) + kW(V) = SC_{2n+1}^k(A) + kW(U) + kW(V)$$

Thus, we observe that $W(UV) = W(U) + W(V)$, since k can be any integer.

In three dimensions, $W(UV) = W(U) + W(V)$ can be easily proved by straightforward derivation, which is presented in the following.

The explicit expression of the winding number is $W(U) = -\frac{1}{24\pi^2} \int_{T^3} \text{Tr}(UdU^\dagger)^3$. Then, for UV , we have

$$UVd(V^\dagger U^\dagger) = U(VdV^\dagger + dU^\dagger U)U^\dagger.$$

The integrand can be expanded as

$$\begin{aligned} \text{Tr}[UVd(V^\dagger U^\dagger)]^3 &= \text{Tr}(UdU^\dagger)^3 + \text{Tr}(VdV^\dagger)^3 \\ &\quad + 3\text{Tr}(VdV^\dagger VdV^\dagger dU^\dagger U) + 3\text{Tr}(VdV^\dagger dU^\dagger U dU^\dagger U) \\ &= \text{Tr}(UdU^\dagger)^3 + \text{Tr}(VdV^\dagger)^3 \\ &\quad - 3\text{Tr}(dVdV^\dagger dU^\dagger U) - 3\text{Tr}(VdV^\dagger dU^\dagger dU). \end{aligned}$$

Above, we have used the identity $UdU^\dagger U = -dU$. It is observed that the last two terms constitute a total derivative, i.e.,

$$d\text{Tr}(VdV^\dagger dU^\dagger U) = \text{Tr}(dVdV^\dagger dU^\dagger U) + \text{Tr}(VdV^\dagger dU^\dagger dU).$$

Thus, on a compact manifold, the two terms do not contribute to the integral, and therefore we have $W(UV) = W(U) + W(V)$.

In the revised manuscript, we have added relevant discussions in the newly added section of Topological invariants in Methods and cited Nakahara's book.

Reviewer #2 (Remarks on Code):

Comment: *The codes are clear and simple. It is convincing that the results are reproducible.*

Our reply: We sincerely appreciate the referee's recognition and acknowledgment of the correctness of our codes.

Reviewers' Comments:

Reviewer #1:

Remarks to the Author:

With much interest I have studied the resubmitted version of the manuscript by R.Xiao and Y.X.Zhao on "Revealing the spatial nature of sublattice symmetry". I am glad to report that the authors have adequately responded to all my raised comments. Transferring the mathematical details and derivations to the Methods section has significantly improved the presentation of the theoretical findings and of the overall value of this work. The manuscript is clearly written and well structured, and the presented results will find immediate applications in designing and understanding properties of topological metamaterials.

I have no further questions and support the publication of the manuscript in Nature Communications.

Minor comments:

-- on p5, in sentence "Instead of the half BZ, we just apply all the rationale below on the whole BZ." It is not clear what "below" refers to, as this the sentence appears at the end of the main text.

-- on p7, between Eqs.(26-27), in "Since H^{d+1} is constant at $k=0$ and $k=\pi$ ", it seems to me that k should be replaced by θ .

-- on p7, in Eq.(30), "d" appears in superscripts in two different meanings: as dimension of the momentum space (on the left), and for the "doubled cell convention" (on the right). Perhaps less confusing notation is possible, e.g. by using capital D (or another font) for "dimension".

-- on p8, in the opening of paragraph on "Bulk-boundary correspondence", it may be helpful to emphasize that the presented geometric picture of for the Z_2 invariant applies specifically to the two-dimensional case.

-- on p8, above Eq.(51): Stoke's  Stokes'.

Reviewer #2:

Remarks to the Author:

Second review of the manuscript NCOMMS-23-59837

'Revealing the spatial nature of sublattice symmetry'

by Rong Xiao and Y. X. Zhao

I am satisfied with the reply of the authors and revisions made in the text.

I only have one small remark concerning bulk-boundary correspondence, the sentence:

"But, the quantized Berry phase n of the 1D ky-subsystem with $k_x = k_0x$ implies the existence of an in-gap mode at each end"

It was shown, e.g., in work Phys. Rev. B 95, 035421 (2017) that quantized Zak phase not necessary lead to presence of the end-states. It may easily happen that it just leads to charge accumulation.

If authors could clarify this then I would have no further comments and my recommendation would be to publish this work in Nature Communications.

Response to referee reports

-----Response to Report of Referee #1 -----

Referee comment: *on p5, in sentence "Instead of the half BZ, we just apply all the rationale below on the whole BZ." It is not clear what "below" refers to, as this the sentence appears at the end of the main text.*

Our reply: Thank you for raising the issue of this confusing statement. We have removed the word “below” to avoid confusion.

Referee comment: *on p7, between Eqs.(26-27), in "Since H^{d+1} is constant at $k=0$ and $k=\pi$ ", it seems to me that k should be replaced by θ_d .*

Our reply: Thank you for pointing this mistake out. We have replaced k by θ_d in the revised manuscript.

Referee comment: *on p7, in Eq.(30), "d" appears in superscripts in two different meanings: as dimension of the momentum space (on the left), and for the "doubled cell convention" (on the right). Perhaps less confusing notation is possible, e.g. by using capital D (or another font) for "dimension".*

Our reply: Thank you for raising the issue. To avoid this confusion, we have added an appropriate comment, that is, the superscript “d” denotes the dimension of the momentum space, and “(d)” indicates the “double unit cell convention”.

Referee comment: *on p8, in the opening of paragraph on "Bulk-boundary correspondence", it may be helpful to emphasize that the presented geometric picture of for the Z_2 invariant applies specifically to the two-dimensional case.*

Our reply: Thank you for raising the issue. We have mentioned this point in the opening of the paragraph in the revised manuscript.

Referee comment: *on p8, above Eq.(51): Stoke's  Stokes'.*

Our reply: Thank you for pointing this typo out. We have corrected it in the revised manuscript.

-----Response to Report of Referee #2 -----

Referee comment: *I only have one small remark concerning bulk-boundary correspondence, the sentence: "But, the quantized Berry phase π of the 1D k_y -subsystem with $k_x = k_0x$ implies the existence of an in-gap mode at each end." It was shown, e.g., in work Phys. Rev. B 95, 035421 (2017) that quantized Zak phase not*

necessary lead to presence of the end-states. It may easily happen that it just leads to charge accumulation.

Our reply: Thank you for raising the issue.

The PRB paper studies the Zak phase quantized by inversion or spacetime inversion symmetry. The failure of the bulk-boundary correspondence of the Zak phase occurs only in the case of so-called “intra-cellular” inversion symmetry. In this case, the symmetry and the unit cell are not compatible for finite systems, i.e., if a finite system preserves inversion symmetry, the system is not an integer multiple of unit cells, because the inversion center is not the unit-cell center.

In our work, we assume that the unit cells are compatible with protecting symmetries, and therefore the bulk-boundary correspondence of the Zak phase still holds.

Reviewer #2 (Remarks on Code):

Comment: *The codes are clear and simple. It is convincing that the results are reproducible.*

Our reply: We sincerely appreciate the referee’s recognition and acknowledgment of the correctness of our codes.